# Neutrophil extracellular traposis in cancer patients with acute ischemic stroke

**Jae-Won Hyun**[1]*, **Rosah May Palermo Payumo**[1], **Jieun Chung**[1], **You-Ri Kang**[1], **Su-Hyun Kim**[1], **Ho Jin Kim**[1], **Ji-Youn Han**[2], **Sang-Yoon Park**[3]

1 Department of Neurology, Research Institute and Hospital of National Cancer Center, Goyang, Republic of Korea, 2 Center for Lung Cancer, Research Institute and Hospital of National Cancer Center, Goyang, Republic of Korea, 3 Center for Gynecologic Cancer, Research Institute and Hospital of National Cancer Center, Goyang, Republic of Korea

* jacksy12@naver.com

## Abstract

### Background

Patients with cancer exhibit an increased risk of acute ischemic stroke (AIS), and neutrophil extracellular traposis (NETosis) has been proposed as a mechanism underlying cancer-associated hypercoagulability. However, studies validating these findings in independent cohorts are limited.

### Objective

We sought to explore whether NETosis-associated markers (plasma DNA and nucleosomes) are increased in patients with active cancer and AIS, and whether these increases correlate with coagulopathy markers in cancer patients.

### Methods

We analyzed NETosis-associated markers in cancer patients with and without AIS and healthy controls, and assessed the correlation between these markers and coagulopathy markers. Additionally, we compared the levels of Netosis-associated markers between cancer patients with conventional stroke mechanisms (CSM) and those with embolic stroke of undetermined source (ESUS).

### Results

Plasma DNA and nucleosome levels were significantly higher in cancer patients with AIS than in cancer controls and healthy controls (p < 0.001, respectively). Both markers correlated with D-dimer levels in cancer patients with AIS. In a sub-analysis, cancer patients with ESUS showed higher levels of NETosis-associated markers compared to those with CSM, whereas vascular risk factors were more frequently observed in cancer patients with CSM.

**Data availability statement:** All relevant data are within the paper and its Supporting Information files.

**Funding:** This study was supported by National Cancer Center in Korea (Grant No. 2410550 & 2510250). The funders had no role in study design, data collection and analysis, decision to publish, or preparation of the manuscript.

**Competing interests:** YRK, RMPP, JC, SYP report no financial disclosures. JWH has received a grant from the National Cancer Center. SHK has lectured, consulted, and received honoraria from Bayer Schering Pharma, Biogen, Genzyme, Merck Serono, and UCB and received a grant from the National Cancer Center. HJK received a grant from the National Research Foundation of Korea and research support from Aprilbio, Eisai, Good T cells and UCB; received consultancy/speaker fees from Alexion, Altos Biologics, Astra Zeneca, Biogen, Daewoong Pharmaceutical, Eisai, GC Pharma, Handok Pharmaceutical, Kaigene, Kolon Life Science, MDimune, Merck, Mitsubishi Tanabe Pharma, Roche, and Sanofi; is a co-editor for the Multiple Sclerosis Journal and an associated editor for the Journal of Clinical Neurology. JYH received consulting fees from Astro Zeneca, Abbvie, LG Chem, Dae Woong, Lantern, Jassen, Takeda, Roche, Amgen, Daiichi Sankyo, Oncovix, Novartis, BMS, Merck, Pfizer; received honoraria from Astra Zeneca, Takeda, Norvatis, Pfized, Takeda, Jassen, Merck, Yuhan, Roche; received payment for expert testimony from Astra Zeneca; participated on a data safety monitoring board from Jassen, Astra Zeneca. This does not alter our adherence to PLOS ONE policies on sharing data and materials.

## Conclusion

These findings suggest that NETosis may contribute to hypercoagulability in patients with active cancer and AIS, particularly in those with ESUS. These results provide additional evidence supporting the establishment of pathophysiology-based therapeutic approaches.

## Introduction

Cancer and acute ischemic stroke (AIS) are among the leading causes of mortality in the older adult population [1–2]. The association between cancer and an increased risk of AIS has been observed [1–2]. However, the pathogenesis of cancer-associated AIS remains incompletely understood, and standardized therapeutic and preventive strategies for patients with cancer-associated AIS have yet to be established. Neutrophil extracellular traposis (NETosis) is an inflammatory process characterized by neutrophil cell death and the release of extracellular DNA webs [3]. NETosis results in the release of nuclear components such as cell-free DNA and nucleosomes into the circulation, which can be quantified as surrogate markers of NET formation [3]. A previous study has proposed NETosis as a mechanism potentially contributing to hypercoagulability in cancer patients [4]. However, investigations validating these results in other independent cohorts are scarce.

In the present study, we aimed to explore whether NETosis-associated markers are elevated in patients with active cancer and AIS, and whether these elevations are associated with coagulopathic markers in cancer patients.

## Materials and methods

Patients with active cancer who 1) experienced AIS and 2) had plasma samples stored within 1 month after AIS in the biobank of the National Cancer Center (NCC), from 13th Jan 2020–25th Oct 2023, were included in this study. AIS was defined as sudden onset neurological deterioration accompanied by relevant lesions in magnetic resonance imaging (MRI) with diffusion restriction. Plasma samples from the NCC bio-bank were primarily collected at the first visit to the NCC, usually at the time of cancer diagnosis, before the initiation of treatment, and stored at −80 °C until analysis. Active cancer was defined as a diagnosis of cancer at the time of sampling [2]. Age-/sex-/primary tumor types-/presence of distant metastasis-matched cancer controls but without AIS, who had stored plasma samples in the NCC bio-bank from 15th Jun 2020–20th Feb 2024, were also included. Additionally, age- and sex-matched healthy controls with stored plasma samples in the NCC bio-bank from 7th Nov 2007 to 31st Oct 2023, but without known vascular risk factors and cancer history, were included.

We excluded patients who 1) had insufficient information for the evaluation of conventional stroke mechanisms including results of cerebro-vascular imaging and/or cardiac assessment, 2) were treated with antithrombotic agents or cancer chemotherapy at the time of sampling for NETosis-associated markers to minimize treatment

effects, 3) had known recent infection history to exclude other provocation factor for NETosis, and 4) had hematologic malignancies. The clinical, laboratory, and radiological data of enrolled participants were systematically reviewed. The data were accessed for research purposes from 4th April 2024–30th Oct 2024.

NETosis-associated markers, including plasma DNA and nucleosome, were assessed by blinded investigators. Plasma DNA was extracted using the QIAamp Circulating Nucleic Acid Kit (Qiagen) and quantified using the Qubit dsDNA Quantification Assay (Invitrogen), while nucleosomes were analyzed using the Cell Death Detection ELISA Kit (Roche), according to the manufacturers' protocols.

We compared the levels of NETosis-associated markers (plasma DNA and nucleosomes) among cancer patients with AIS, cancer patients without AIS, and healthy controls to determine the association between NET formation and AIS in patients with active cancer [4], Additionally, we examined the correlation between the levels of D-dimer, a known marker of coagulopathy [5], and NETosis-associated markers to evaluate the association between NETosis and hypercoagulopathy. Subsequently, we compared the levels of NETosis-associated markers between cancer patients with AIS who had conventional stroke mechanisms (CSM), including large-artery atherosclerosis, cardio-embolism, or lacunar stroke, and those with embolic stroke of undetermined source (ESUS) [6], to investigate potential cancer-associated stroke mechanisms. Finally, we evaluated the correlation between the levels of D-dimer and NETosis-associated markers in the ESUS group alone.

The differences between cancer patients with AIS and the control groups were compared using the Kruskal–Wallis test for continuous variables and the chi-square test for categorical variables. Correlations between D-dimer levels and NETosis-associated markers were analyzed using Spearman analysis. Fisher's exact test for categorical variables and the Mann–Whitney U test for continuous variables were performed to compare cancer patients with AIS, with and without CSM. A p-value < 0.05 was considered statistically significant. This study was approved by the Institutional Review Board of the NCC (No. NCC 2024−0098), and we had access to de-identified information during or after data collection.

## Results

Plasma DNA and nucleosome levels were analyzed in 50 and 40 cancer patients with AIS, respectively. In 56 cancer controls (cancer patients without AIS) and 60 healthy controls, both plasma DNA and nucleosome levels were assessed. The characteristics of the enrolled participants are presented in Table 1. There were no significant differences in age or sex between cancer patients with AIS and the control groups, nor were there significant differences in vascular risk factors, primary tumor types and presence of distant metastasis between cancer patients with AIS and cancer controls.

Plasma DNA and nucleosome levels were significantly higher in cancer patients with AIS than in the control groups (p < 0.001 for all comparisons, Table 1 & Fig 1). The plasma DNA and nucleosome levels significantly correlated with D-dimer levels (r = 0.5214, p < 0.001 for plasma DNA; r = 0.5015, p = 0.001 for nucleosomes; Fig 2A & 2B). In 36 cancer patients with AIS with available data for both plasma DNA and nucleosome levels, a significant correlation was observed between the two markers (r = 0.9406, p < 0.001; Fig 2C).

In the sub-analysis of cancer patients with AIS, comparing those with ESUS to those with CSM, plasma DNA levels were measured in 37 patients with ESUS and 13 patients with CSM, while nucleosome levels were assessed in 30 patients with ESUS and 10 patients with CSM (Table 2). The levels of NETosis-associated markers were significantly higher in patients with ESUS than in those with CSM (p < 0.001 for plasma DNA, p = 0.0032 for nucleosomes; Fig 3). In contrast, vascular risk factors including diabetes mellitus, hypertension, and dyslipidemia were more frequently observed in cancer patients with CSM than in those with ESUS. No significant differences in the time from AIS onset to sampling were observed between the ESUS and CSM groups for plasma DNA or nucleosome.

In the ESUS group, both plasma DNA and nucleosome levels showed a significant correlation with D-dimer levels (r = 0.3273, p = 0.048 for plasma DNA; r = 0.409, p = 0.025 for nucleosomes; Fig 4A and 4B). Among 26 cancer patients with ESUS for whom both plasma DNA and nucleosome levels were available, a significant correlation was observed between the two markers (r = 0.913, p < 0.001; Fig 4C).

**Table 1. Demographics and characteristics of the entire enrolled participants.**

| Plasma DNA | Cancer associated acute ischemic stroke (n = 50) | Cancer-control (n = 56) | Healthy control (n = 60) | p-value |
|---|---|---|---|---|
| **Age (median [IQR], y)** | 67 [58;74] | 62.5 [57;69] | 63.5 [55;69] | 0.086 |
| **Sex (female)** | 20 (40.0%) | 25 (44.6%) | 31 (51.7%) | 0.463 |
| **Vascular risk factor** | | | | |
| DM | 21 (42.0%) | 19 (33.9%) | – | 0.392 |
| HTN | 29 (58.0%) | 23 (41.1%) | – | 0.082 |
| DL | 15 (30.0%) | 16 (28.6%) | – | 0.872 |
| **Primary cancer** | | | | |
| Lung | 17 (34.0%) | 17 (30.4%) | – | 0.688 |
| Ovary | 11 (22.0%) | 10 (17.9%) | – | 0.593 |
| Others | 22 (44.0%) | 29 (51.8%) | – | 0.423 |
| Presence of distant metastasis | 25 (50.0%) | 23 (41.1%) | – | 0.357 |
| Plasma DNA (median [IQR], ng/ml) | 18.1* [10.0;30.7] | 8.75 [6.53;14.4] | 5.8 [4.6;8.1] | <0.001* |
| Nucleosome | Cancer associated acute ischemic stroke (n = 40) | Cancer-control (n = 56) | Healthy control (n = 60) | p-value |
| **Age (median [IQR], y)** | 64.5 [57.25;70.75] | 62.5 [57;69] | 63.5 [55;69] | 0.700 |
| **Sex (female)** | 18 (45.0%) | 25 (44.6%) | 31 (51.7%) | 0.704 |
| **Vascular risk factor** | | | | |
| DM | 12 (30.0%) | 19 (33.9%) | – | 0.685 |
| HTN | 18 (45.0%) | 23 (41.1%) | – | 0.701 |
| DL | 11 (27.5%) | 16 (28.6%) | – | 0.908 |
| **Primary cancer** | | | | |
| Lung | 16 (40.0%) | 17 (30.4%) | – | 0.327 |
| Ovary | 11 (27.5%) | 10 (17.9%) | – | 0.260 |
| Others | 13 (32.5%) | 29 (51.8%) | – | 0.060 |
| Presence of distant metastasis | 22 (55.0%) | 23 (41.1%) | – | 0.178 |
| Nucleosome (median [IQR], optical density) | 0.655* [0.345;1.161] | 0.22 [0.133;0.358] | 0.11 [0.07;0.158] | <0.001* |

**Abbreviations:** IQR, interquatile rate; y, year; DM, diabetes mellitus; HTN, hypertension; DL, dyslipidemia.

## Discussion

In the current study, active cancer patients with AIS had significantly elevated plasma DNA and nucleosome levels compared to control groups, with these markers correlated with D-dimer levels representing hyper-coagulopathy. Subgroup analysis showed that cancer patients with ESUS had significantly higher levels of NETosis-associated markers correlated with D-dimer levels but a lower frequency of vascular risk factors compared to those with CSM. These findings suggest that Netosis would be one of the pathogenesis in patients with cancer associated AIS, particularly in those with ESUS.

The pathomechanisms of NETosis associated with hypercoagulopathy involve the formation of a scaffold comprising peripheral blood components, such as red blood cells, platelets, and platelet adhesion molecules, as well as the activation of coagulation pathways [7]. The systemic effects of cancer are considered to elevate neutrophil levels and promote NET formation [8]. The preclinical study has shown that breast tumor-derived extracellular vesicles (EVs) injected into mice treated with granulocyte colony-stimulating factor accelerated venous thrombosis, suggesting that cancer-associated thrombosis may be facilitated by both neutrophils and tumor-derived EVs [9]. Clinical data from the OASIS-Cancer study corroborated these findings, demonstrating elevated levels of tumor-derived EVs and

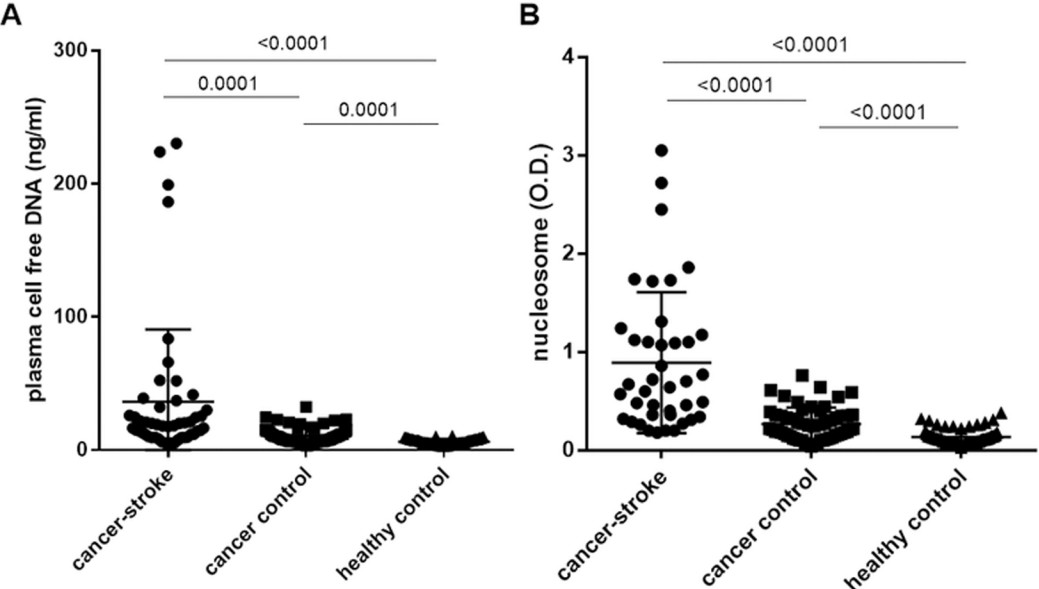

**Fig 1. Comparison of the levels of (A) plasma cell free DNA and (B) nucleosomes among patients with cancer associated acute ischemic stroke and control groups.**

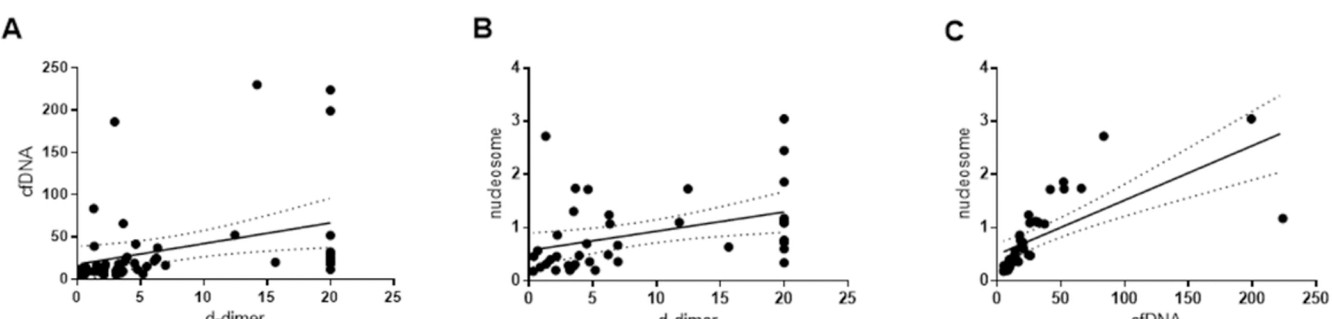

**Fig 2. Correlation of (A) plasma cell free DNA and (B) nucleosome levels with D-dimer level, and (C) correlation of plasma cell free DNA with nucleosome levels.**

NETosis-associated markers in cancer-associated AIS [4,10,11] Collectively, these results suggest a potential role for tumor-derived mechanisms in driving NETosis within cancer-associated AIS. Consistent with previous studies, we reconfirmed the association between NETosis and cancer-associated AIS in an independent cohort of treatment-naïve patients, including those naïve to both chemotherapy for cancer and anti-thrombotic therapy. Our study further highlights the role of NETosis in the pathogenesis of cancer-associated AIS, enhancing our comprehension of its association with hypercoagulopathy.

Understanding the immunological mechanisms underlying cancer-associated AIS carries significant clinical importance, as anti-inflammatory therapies targeting NETosis may offer an additional or alternative preventive strategy to conventional antithrombotic agents, which pose a bleeding risk in cancer patients [12]. Furthermore, recent studies have suggested that NETosis may promote carcinogenesis by facilitating cellular proliferation and angiogenesis [13]. In the current study,

**Table 2. Comparison of cancer-stroke patients with or without conventional stroke mechanism.**

| Plasma DNA | Cancer-stroke patients with ESUS (n = 37) | Cancer-stroke patients with CSM (n = 13) | p-value |
|---|---|---|---|
| Age (median [IQR], y) | 67 [58.5;75] | 65 [57.5;71] | 0.383 |
| Sex (female) | 17 (45.9%) | 3 (23.1%) | 0.197 |
| Vascular risk factor | | | |
| DM | 11 (29.7%) | 10 (76.9%)* | 0.007* |
| HTN | 17 (45.9%) | 12 (92.3%)* | 0.004* |
| DL | 6 (16.2%) | 9 (69.2%)* | 0.001* |
| Primary cancer | | | |
| Lung | 12 (32.4%) | 5 (38.5%) | 0.741 |
| Ovary | 9 (24.3%) | 2 (15.4%) | 0.704 |
| Others | 16 (43.2%) | 6 (46.2%) | 1.000 |
| Presence of distant metastasis | 21 (56.8%) | 4 (30.8%) | 0.196 |
| Plasma DNA (median (IQR), ng/ml) | 21.1* [14.6;38.1] | 9.4 [5.45;14.7] | <0.001* |
| Time to sampling from onset of AIS (median (IQR), week) | 1 [0;2.5] | 1 [0;1.5] | 0.777 |
| **Nucleosome** | **Cancer-stroke patients with ESUS (n = 30)** | **Cancer-stroke patients with CSM (n = 10)** | **p-value** |
| Age (median, y) | 67 [58;73] | 62 [56;65.5] | 0.155 |
| Sex (female) | 15 (50.0%) | 3 (30.0%) | 0.465 |
| Vascular risk factor | | | |
| DM | 6 (20%) | 6 (60.0%)* | 0.041* |
| HTN | 10 (33.3%) | 8 (80.0%)* | 0.025* |
| DL | 5 (16.7%) | 6 (60.0%)* | 0.014* |
| Primary cancer | | | |
| Lung | 11 (36.7%) | 5 (50.0%) | 0.482 |
| Ovary | 9 (30.0%) | 2 (20.0%) | 0.696 |
| Others | 10 (33.3%) | 3 (30.0%) | 1.000 |
| Presence of distant metastasis | 19 (63.3%) | 3 (30.0%) | 0.140 |
| Nucleosome (median (IQR), optical density) | 0.815* [0.445;1.258]* | 0.275 [0.20;0.603] | 0.003* |
| Time to sampling from onset of AIS (median (IQR), week) | 1 [0;2.25] | 1 [0;2] | 0.799 |

**Abbreviations:** ESUS, embolic stroke of undetermined source; CSM, conventional stroke mechanism; IQR, interquatile rate; y, year; DM, diabetes mellitus; HTN, hypertension; DL, dyslipidemia; AIS, acute ischemic stroke.

NETosis-associated markers were highest in cancer patients with AIS but were also elevated in cancer patients without AIS compared to healthy controls. These findings suggest that anti-NETosis agents could be beneficial not only for preventing AIS but also for managing cancer. Some clinical trials are currently investigating anti-inflammatory therapies for AIS of various etiologies [14], and the results of the present study could inform future trials focusing on preventive and therapeutic strategies for cancer-associated AIS.

Owing to the retrospective design based on a single referral center, several unintentional biases may exist, including the high disease severity of the enrolled patients and irregular time intervals from symptom onset to sample collection. However, this design also ensured a relatively uniform evaluation and management of patients, with samples consistently collected and stored under standardized conditions. Future larger prospective multicenter evaluations with longitudinal follow-up are required.

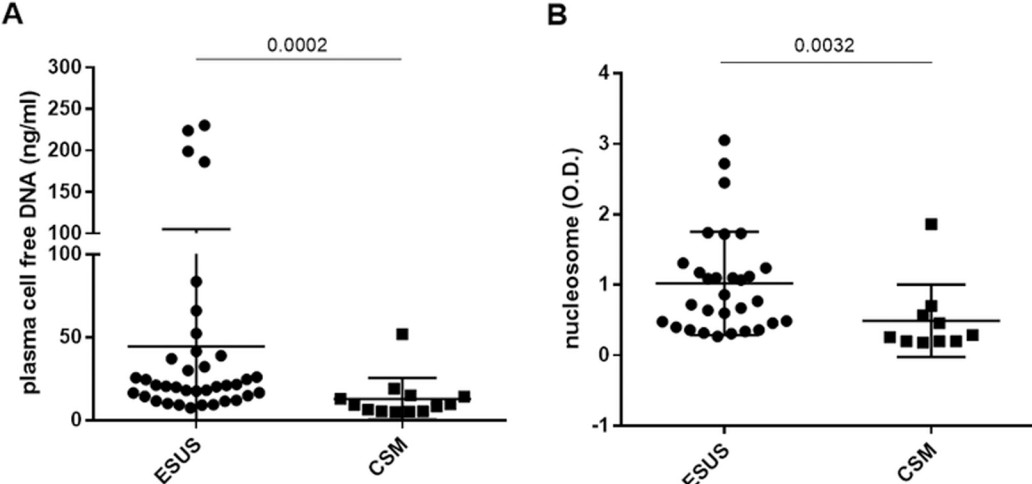

**Fig 3. Comparison of the levels of (A) plasma cell free DNA and (B) nucleosome between cancer patients with embolic stroke of undetermined source (ESUS) and those with conventional stroke mechanism (CSM).**

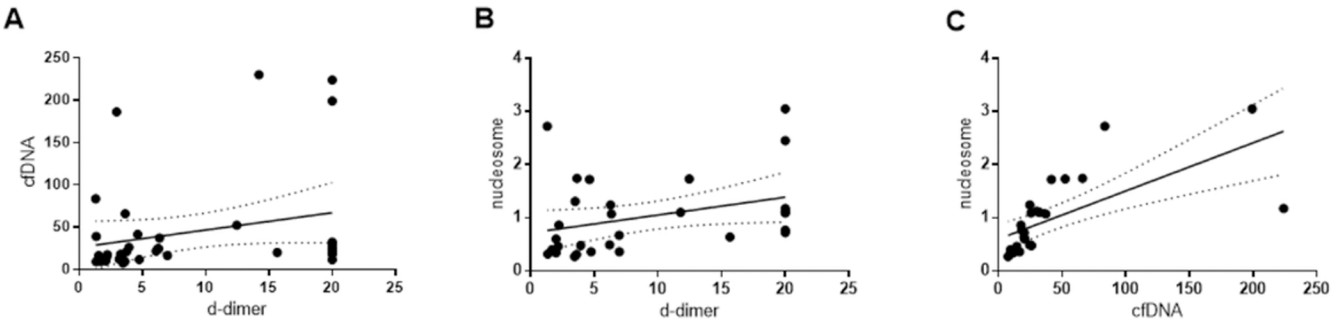

**Fig 4. Correlation of (A) plasma cell free DNA and (B) nucleosome levels with D-dimer level, and (C) correlation of plasma cell free DNA with nucleosome levels in the embolic stroke of undetermined sorce (ESUS) group.**

## Conclusions

In conclusion, NETosis may be one of the underlying mechanisms of hypercoagulopathy in patients with active cancer and AIS, particularly in those with ESUS. These results provide additional evidence to support pathophysiology-driven therapeutic and preventive strategies for managing AIS in patients with cancer.

## Supporting information

**S1. Minimal data set.**
(XLSX)

## Acknowledgments

Blood samples and data were provided NCC Bio Bank of National Cancer Center, Korea. This work was technically supported by the Genomics Core Facility in NCC, Korea.

## Author contributions

**Conceptualization:** Jae-Won Hyun.

**Data curation:** Jae-Won Hyun, Rosah May Palermo Payumo, Jieun Chung, You-Ri Kang, Ji-Youn Han, Sang-Yoon Park.

**Formal analysis:** Jae-Won Hyun.

**Funding acquisition:** Jae-Won Hyun.

**Investigation:** Jae-Won Hyun, Rosah May Palermo Payumo, Jieun Chung.

**Methodology:** Jae-Won Hyun.

**Project administration:** Jae-Won Hyun.

**Resources:** Jae-Won Hyun, Ji-Youn Han, Sang-Yoon Park.

**Supervision:** Jae-Won Hyun.

**Validation:** Jae-Won Hyun, You-Ri Kang, Su-Hyun Kim, Ho Jin Kim, Ji-Youn Han, Sang-Yoon Park.

**Visualization:** Jae-Won Hyun.

**Writing – original draft:** Jae-Won Hyun.

**Writing – review & editing:** Jae-Won Hyun, Su-Hyun Kim, Ho Jin Kim.

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
