## [Decision Letter · Decision Letter 0]

PONE-D-25-06502Neutrophil Extracellular Traposis in Cancer Patients with Acute Ischemic StrokePLOS ONE

Dear Dr. Hyun,

Thank you for submitting your manuscript to PLOS ONE. After careful consideration, we feel that it has merit but does not fully meet PLOS ONE’s publication criteria as it currently stands. Therefore, we invite you to submit a revised version of the manuscript that addresses the points raised during the review process.

We look forward to receiving your revised manuscript.

Kind regards,

Kokouvi Kassegne

Academic Editor

PLOS ONE

Journal Requirements:

“This study was supported by National Cancer Center in Korea (Grant No. 2410550 & 2510250).” 

“YRK, RMPP, JC, SYP report no financial disclosures. JWH has received a grant from the National Cancer Center. SHK has lectured, consulted, and received honoraria from Bayer Schering Pharma, Biogen, Genzyme, Merck Serono, and UCB and received a grant from the National Cancer Center. HJK received a grant from the National Research Foundation of Korea and research support from Aprilbio, Eisai, Good T cells and UCB; received consultancy/speaker fees from Alexion, Altos Biologics, Astra Zeneca, Biogen, Daewoong Pharmaceutical, Eisai, GC Pharma, Handok Pharmaceutical, Kaigene, Kolon Life Science, MDimune, Merck, Mitsubishi Tanabe Pharma, Roche, and Sanofi; is a co-editor for the Multiple Sclerosis Journal and an associated editor for the Journal of Clinical Neurology. JYH received consulting fees from Astro Zeneca, Abbvie, LG Chem, Dae Woong, Lantern, Jassen, Takeda, Roche, Amgen, Daiichi Sankyo, Oncovix, Novartis, BMS, Merck, Pfizer; received honoraria from Astra Zeneca, Takeda, Norvatis, Pfized, Takeda, Jassen, Merck, Yuhan, Roche; received payment for expert testimony from Astra Zeneca; participated on a data safety monitoring board from Jassen, Astra Zeneca.”

Additional Editor Comments:

Please provide point-by-point response to the reviewers' comments and revise the manuscript accordingly, in tracking change mode to show where revisions have been made.

Reviewers' comments:

Reviewer's Responses to Questions

**Comments to the Author**

1. Is the manuscript technically sound, and do the data support the conclusions?

Reviewer #1: Yes

Reviewer #2: Yes

2. Has the statistical analysis been performed appropriately and rigorously? 

Reviewer #1: Yes

Reviewer #2: Yes

3. Have the authors made all data underlying the findings in their manuscript fully available?

Reviewer #1: Yes

Reviewer #2: Yes

4. Is the manuscript presented in an intelligible fashion and written in standard English?

Reviewer #1: Yes

Reviewer #2: Yes

5. Review Comments to the Author

Reviewer #1: “AIS was defined as sudden onset neurological deterioration accompanied by relevant lesions in magnetic resonance imaging (MRI) with diffusion restriction” – metastasis and other causes can also have diffusion restrictions but are not ischemic stroke.

One of the exclusion criteria is “patients treated with antithrombotic agents or cancer chemotherapy at the time of sampling for NET-associated markers to minimize treatment effects”. If these patients are a month from stroke onset, most will have antiplatelets started.

Similar to the point above, no chemotherapy or antithrombotics would mean that this cohort may be sub-selected to only include pallative cases or very sick patients.

This difficult inclusion and exclusion criteria may explain the small number of patients recruited after 4 years in a national level database.

Was there any association between D-dimer and plasma free DNA level and nucleosome levels in the ESUS cohort alone?

Reviewer #2: The authors are to be congratulated for this well-written manuscript, which validates the results of the OASIS Study. The latter demonstrated an association between Neutrophil Extracellular Traposis Biomarkers and active cancers in patients with AIS.

Below are a few minor comments regarding the general comprehension of the project, the description of the method and the reporting of results.

1) According to the journal ‘’Statistical Reporting’’ please report p-values less than 0.001 as p < 0.001. In the abstract, in the manuscript and in tables/figures we can find p < 0.0001 as well as higher p values up to the fourth decimal place, which is not usual.

2) Authors must clearly define what was the definition of “active cancer” used in their study. The source provided does not clearly define 'active cancer'. Was the definition proposed by the International Society on Thrombosis and Haemostasis used for identification of active cancer? What about cancers diagnosed in the year following the AIS (so-called occult cancer)? Were they excluded or considered as active cancer, as it is something the case?

3) The authors mentioned the presence of relevant lesions in magnetic resonance imaging (MRI) with diffusion restriction as inclusion criteria. Were patients who only received a brain CT excluded from the study? If so, please specify this in the methods.

4) It would be useful for the readers to have more background on the context/rationale explaining why plasma DNA and nucleosome are NETosis-associated markers. Please develop on it in the introduction or discussion.

5) In the first line of the results, please specify that it is 50 and 40 "cancer" patients with AIS.

6) In the description of the statistical analyses in the method, please specify how the results are reported/determined significant: r and p-value with threshold used.

6. PLOS authors have the option to publish the peer review history of their article (what does this mean?). If published, this will include your full peer review and any attached files.

Reviewer #1: **Yes: **Leonard Yeo

Reviewer #2: **Yes: **Morin Beyeler, MD, PhD

---

## [Author Response · Author response to Decision Letter 1]

13 May 2025

Dear Professor Kokouvi Kassegne,

We wish to re-submit the manuscript titled “Neutrophil Extracellular Traposis in Cancer Patients with Acute Ischemic Stroke”. We thank the reviewers for their thoughtful suggestions. Herein, we respond to the reviewers’ comments and hope that we have adequately addressed their concerns. All the changes in manuscript are highlighted with track changes.

Thank you for further consideration of our manuscript. We look forward to hearing from you.

Sincerely,

Jae-Won Hyun

jacksy12@naver.com

Reviewer #1: “AIS was defined as sudden onset neurological deterioration accompanied by relevant lesions in magnetic resonance imaging (MRI) with diffusion restriction” – metastasis and other causes can also have diffusion restrictions but are not ischemic stroke.

While we recognize that diffusion restriction on MRI is not specific to ischemic stroke, we defined AIS as “sudden onset” neurological deterioration with corresponding diffusion-restricted lesions to reflect the acute clinical course characteristic of vascular events. This distinction may help differentiate AIS from other etiologies such as metastasis, which generally present with a more progressive clinical evolution.

One of the exclusion criteria is “patients treated with antithrombotic agents or cancer chemotherapy at the time of sampling for NET-associated markers to minimize treatment effects”. If these patients are a month from stroke onset, most will have antiplatelets started. Similar to the point above, no chemotherapy or antithrombotics would mean that this cohort may be sub-selected to only include pallative cases or very sick patients. This difficult inclusion and exclusion criteria may explain the small number of patients recruited after 4 years in a national level database.

We appreciate the opportunity to clarify this point. As described in the Methods section, plasma samples from the NCC bio-bank were collected at the first visit to the NCC, before the initiation of treatment. Given the increased risk of AIS around the time of cancer diagnosis (Wang Y et al, Front Neurol 2019;10:579), many patients experienced AIS shortly after sample collection, reflecting early-stage rather than palliative cases. As NETosis markers may potentially be affected by treatment, patients who had AIS before cancer diagnosis and had already started antithrombotics elsewhere were excluded. While this may have slightly limited the sample size, such cases were uncommon. Similarly, patients referred after initiating cancer treatment at outside hospitals were also excluded, though infrequent.

To more clearly reflect the timing and context of sampling, we have added the phrases “primarily” and “usually” to the Methods section: “Plasma samples from the NCC bio-bank were primarily collected at the first visit to the NCC, usually at the time of cancer diagnosis, before the initiation of treatment, and stored at −80 °C until analysis.”

Was there any association between D-dimer and plasma free DNA level and nucleosome levels in the ESUS cohort alone?

Thank you for your insightful comments. We conducted an additional analysis to assess the association between D-dimer levels and NETosis-associated markers in the ESUS group and found significant correlations. The following information has been added to the Methods, Results (Figure 4), and Discussions.

Methods: “Finally, we evaluated the correlation between D-dimer levels and NETosis-associated markers in the ESUS group alone.” Results: “In the ESUS group, both plasma DNA and nucleosome levels showed a significant correlation with D-dimer levels (r = 0.3273, p = 0.048 for plasma DNA; r = 0.409, p = 0.025 for nucleosomes; Figure 4A and B). Among 26 cancer patients with ESUS for whom both plasma DNA and nucleosome levels were available, a significant correlation was observed between the two markers (r = 0.913, p < 0.001; Figure 4C).” Discussions: Subgroup analysis showed that cancer patients with ESUS had significantly higher levels of NETosis-associated markers “correlated with D-dimer levels”… In conclusion, NETosis may be one of the underlying mechanisms of hypercoagulopathy in patients with active cancer and AIS, “particularly in those with ESUS.”

Reviewer #2: The authors are to be congratulated for this well-written manuscript, which validates the results of the OASIS Study. The latter demonstrated an association between Neutrophil Extracellular Traposis Biomarkers and active cancers in patients with AIS. Below are a few minor comments regarding the general comprehension of the project, the description of the method and the reporting of results.

We sincerely thank the reviewer for the encouraging feedback and thoughtful comments regarding our manuscript.

1) According to the journal ‘’Statistical Reporting’’ please report p-values less than 0.001 as p < 0.001. In the abstract, in the manuscript and in tables/figures we can find p < 0.0001 as well as higher p values up to the fourth decimal place, which is not usual.

Thank you for your meticulous comments. We revised the p-values in manuscript and table as you recommended (p < 0.001 and p-values up to the third decimal place).

2) Authors must clearly define what was the definition of “active cancer” used in their study. The source provided does not clearly define 'active cancer'. Was the definition proposed by the International Society on Thrombosis and Haemostasis used for identification of active cancer? What about cancers diagnosed in the year following the AIS (so-called occult cancer)? Were they excluded or considered as active cancer, as it is something the case?

We appreciate the opportunity to clarify this point. Although active cancer is commonly defined as cancer at diagnosis, during active treatment, or at the presence of recurrent or metastatic disease according to the International Society on Thrombosis and Haemostasis, in our study, the definition was simplified. This was because blood samples were primarily collected at the first visit for initial cancer work-up, and patients already undergoing active treatment or with recurrent/metastatic cancer usually after prior cancer treatment were excluded to minimize treatment-related effects on NETosis markers. Therefore, in the Methods section, we defined active cancer as follows: “Active cancer was defined as a diagnosis of cancer at the time of sampling.”

As the National Cancer Center is primarily a cancer referral center and not an acute stroke center, cases of occult cancer presenting after AIS were rare. Moreover, patients referred after AIS diagnosis who had already started antithrombotic treatment at outside hospitals were excluded to avoid potential confounding effects. Therefore, occult cancer was not identified in any of the enrolled patients.

3) The authors mentioned the presence of relevant lesions in magnetic resonance imaging (MRI) with diffusion restriction as inclusion criteria. Were patients who only received a brain CT excluded from the study? If so, please specify this in the methods.

As most biobank samples were collected at the time of the first visit, patients were generally in the early phase of cancer diagnosis and tended to undergo comprehensive evaluations, such as brain MRI, when AIS was clinically suspected. Fortunately, in Korea, access to MRI is relatively high, and the majority of patients with suspected AIS were able to undergo MRI and therefore, no patients were excluded solely because only a brain CT was performed.

4) It would be useful for the readers to have more background on the context/rationale explaining why plasma DNA and nucleosome are NETosis-associated markers. Please develop on it in the introduction or discussion.

Thank you for your meaningful comment. We agreed with you and added the content in introduction as follows: “NETosis results in the release of nuclear components such as cell-free DNA and nucleosomes into the circulation, which can be quantified as surrogate markers of NET formation.”

5) In the first line of the results, please specify that it is 50 and 40 "cancer" patients with AIS.

As you recommended, we added “cancer” in manuscript.

6) In the description of the statistical analyses in the method, please specify how the results are reported/determined significant: r and p-value with threshold used.

We reported correlation results using Spearman’s correlation coefficient (r) and the corresponding p-value. Statistical significance was determined using a p-value < 0.05. Given the limited sample size, we interpreted the findings primarily based on the statistical significance of the p-value, rather than applying conventional thresholds for the magnitude of r, which may be less reliable in small samples (Schober P, et al. Anesth Analg. 2018;126:1763–1768). Accordingly, the following statement was added to the Methods section: “A p-value < 0.05 was considered statistically significant.”

---

## [Decision Letter · Decision Letter 1]

Neutrophil Extracellular Traposis in Cancer Patients with Acute Ischemic Stroke

PONE-D-25-06502R1

Dear Dr. Hyun,

We’re pleased to inform you that your manuscript has been judged scientifically suitable for publication and will be formally accepted for publication once it meets all outstanding technical requirements.

Kind regards,

Kokouvi Kassegne

Academic Editor

PLOS ONE

Additional Editor Comments (optional):

The manuscript is now suitable for publication.

Reviewers' comments:

Reviewer's Responses to Questions

**Comments to the Author**

1. If the authors have adequately addressed your comments raised in a previous round of review and you feel that this manuscript is now acceptable for publication, you may indicate that here to bypass the “Comments to the Author” section, enter your conflict of interest statement in the “Confidential to Editor” section, and submit your "Accept" recommendation.

Reviewer #2: All comments have been addressed

2. Is the manuscript technically sound, and do the data support the conclusions?

Reviewer #2: Yes

3. Has the statistical analysis been performed appropriately and rigorously? 

Reviewer #2: Yes

4. Have the authors made all data underlying the findings in their manuscript fully available?

Reviewer #2: Yes

5. Is the manuscript presented in an intelligible fashion and written in standard English?

Reviewer #2: Yes

6. Review Comments to the Author

Reviewer #2: All my comments have been satisfactorily addressed by the authors and I have no further points before a decision.

7. PLOS authors have the option to publish the peer review history of their article (what does this mean?). If published, this will include your full peer review and any attached files.

Reviewer #2: **Yes: **Morin Beyeler, MD, PhD

---

## [Editor Report · Acceptance letter]

PONE-D-25-06502R1

PLOS ONE

Dear Dr. Hyun,

I'm pleased to inform you that your manuscript has been deemed suitable for publication in PLOS ONE. Congratulations! Your manuscript is now being handed over to our production team.

Kind regards,

on behalf of

Dr. Kokouvi Kassegne

Academic Editor

PLOS ONE